# Permanent Pro-Tumorigenic Shift in Adipose Tissue-Derived Mesenchymal Stromal Cells Induced by Breast Malignancy

**DOI:** 10.3390/cells9020480

**Published:** 2020-02-19

**Authors:** Jana Plava, Marina Cihova, Monika Burikova, Martin Bohac, Marian Adamkov, Slavka Drahosova, Dominika Rusnakova, Daniel Pindak, Marian Karaba, Jan Simo, Michal Mego, Lubos Danisovic, Lucia Kucerova, Svetlana Miklikova

**Affiliations:** 1Cancer Research Institute, Biomedical Research Center, University Science Park for Biomedicine, Slovak Academy of Sciences, 845 05 Bratislava, Slovakia; jana.plava@savba.sk (J.P.); marina.cihova@savba.sk (M.C.); monika.burikova@savba.sk (M.B.); 22nd Department of Oncology, Faculty of Medicine, Comenius University, National Cancer Institute, Klenova 1, 833 10 Bratislava, Slovakia; bohac.md@gmail.com (M.B.); misomego@gmail.com (M.M.); 3Department of Oncosurgery, National Cancer Institute, Klenova 1, 833 10 Bratislava, Slovakia; lubos.danisovic@fmed.uniba.sk; 4Regenmed Ltd., Medena 29, 811 08 Bratislava, Slovakia; 5Comenius University Bratislava, Jessenius Faculty of Medicine Martin, Department of Histology and Embryology, 036 01 Martin, Slovakia; adamkov@jfmed.uniba.sk; 6Hermes LabSystems, s.r.o., 831 06 Bratislava, Slovakia; slavka.drahosova@hermeslab.sk; 7Institute of Medical Biology, Genetics and Clinical Genetics, Faculty of Medicine, Comenius University, 813 72 Bratislava, Slovakia; domeenica@gmail.com (D.R.); daniel.pindak@nou.sk (D.P.); marian.karaba@nou.sk (M.K.); jan.simo@nou.sk (J.S.)

**Keywords:** mesenchymal stromal cells, adipose tissue, breast cancer, tumor microenvironment, perineural invasion

## Abstract

During cancer progression, breast tumor cells interact with adjacent adipose tissue, which has been shown to be engaged in cancer aggressiveness. However, the tumor-directed changes in adipose tissue-resident stromal cells affected by the tumor–stroma communication are still poorly understood. The acquired changes might remain in the tissue even after tumor removal and may contribute to tumor relapse. We investigated functional properties (migratory capacity, expression and secretion profile) of mesenchymal stromal cells isolated from healthy (*n* = 9) and tumor-distant breast adipose tissue (*n* = 32). Cancer patient-derived mesenchymal stromal cells (MSCs) (MSC-CA) exhibited a significantly disarranged secretion profile and proliferation potential. Co-culture with MDA-MB-231, T47D and JIMT-1, representing different subtypes of breast cancer, was used to analyze the effect of MSCs on proliferation, invasion and tumorigenicity. The MSC-CA enhanced tumorigenicity and altered xenograft composition in immunodeficient mice. Histological analysis revealed collective cell invasion with a specific invasive front of EMT-positive tumor cells as well as invasion of cancer cells to the nerve-surrounding space. This study identifies that adipose tissue-derived mesenchymal stromal cells are primed and permanently altered by tumor presence in breast tissue and have the potential to increase tumor cell invasive ability through the activation of epithelial-to-mesenchymal transition in tumor cells.

## 1. Introduction

The interaction of breast epithelium and stroma promotes normal breast structure and function [1]. However, adipose tissue surrounding breast tumors is no longer recognized as a passive structural element, but as a key component contributing to breast cancer progression [2,3]. Diverse stromal cells, including myofibroblasts, pericytes, endothelial cells and cancer-associated fibroblasts (CAF), are recruited from adjacent adipose tissue, and these become an integral component of the tumor microenvironment [4,5]. Mesenchymal stromal cells (MSCs) also migrate to tumors, which are commonly perceived as wounds that never heal [6]. Several studies have elucidated MSC ability to transform into a carcinoma-associated fibroblast phenotype when treated in vitro with breast cancer cell-secreted factors [7,8,9]. This change is demonstrated by a higher expression of αSma, Vimentin, fibroblast specific protein 1 (FSP-1), stromal-derived factor 1 (SDF1) and C-C motif chemokine ligand 5 (CCL5) [10,11]. At least five tumor-associated stromal fibroblast cell subtypes have been identified in the tumor stroma, and these are differentiated by the expression of specific markers associated with various levels of tumor aggression. The most aggressive subtype is characterized by extensive matrix remodeling and the increased expression of FSP-1 and fibroblast activating protein (FAP) [12]. We previously established that factors secreted by tumor cells alter the MSCs’ molecular traits and angiogenic ability. We also confirmed that these changes correlate with their subsequent pro-tumorigenic action and that chemotherapeutically pre-treated MSCs produce a similar effect [13,14]. Additionally, Yeh et al. associated the CXCL1 secreted by stromal cells in breast adipose tissue with doxorubicin resistance mediated by ABCG2 up-regulation [15]. Although studies provide evidence that in vitro MSCs are prone to differentiation into carcinoma-associated fibroblasts induced by tumor secretome [16], there is no experimental evidence as to whether breast malignancy affects the MSCs in distant adipose tissue remaining in the breast after surgery and if those changes may persist even after the tumor removal. Since has also been suggested that interactions between tumor and normal tissue are bi-directional [17,18], we hypothesize that malignant cells and their secretome shape the normal distant tissue and alter its properties.

Herein, we performed a detailed comparison of the functional properties of MSCs from the following four origins in healthy donors and breast cancer patients to determine the influence of malignancy on normal stromal precursors: (1) the first MSC group was isolated from the breast adipose tissue of healthy donors undergoing planned aesthetic breast surgery; (2) the second group comprised MSCs from cancer patient breast adipose tissue adjacent to pre-malignant lesions; (3) the third group was MSCs obtained from adipose tissue of breast cancer patients diagnosed with invasive tumor type and (4) this group contained tissue similar to that in the third group but also harboring the BRCA gene mutation. The results establish that breast cancer patient-derived MSCs are inherently altered; they promote in vivo tumor growth and they also increase tumor cell invasiveness. Finally, identification of key functional changes in the MSCs located in the tumor micro-environment and increased understanding of the mechanisms involved in MSC-enhanced tumor growth and metastasis could initiate new methods of normalizing tumor micro-environments, and thus regulate disease progression.

## 2. Materials and Methods

### 2.1. Cell Cultures

Mesenchymal stromal cells were isolated from the breast adipose tissue of four different donor groups: Group No.1 MSC-H (*n* = 9), isolated from breast adipose tissue of healthy donors, Group No.2 MSC-DCIS (*n* = 2), isolated from adipose tissue adjacent to pre-malignant lesions, Group No.3 MSC-CA (*n* = 24), isolated from adipose tissue adjacent to malignant lesions, and Group No.4 MSC-BRCA+ (*n* = 6), isolated from adipose tissue adjacent to malignant lesions harboring the BRCA gene mutation. We used tumor-adjacent adipose tissue obtained during surgery, and the samples ranged from 1.5 to 5 cm^3^. All donors provided informed consent and all procedures were approved by the Ethics Committee of the Ruzinov University Hospital and the National Cancer Institute (TRUSK-003). The MSCs were isolated as previously described [19]. The isolated cells were maintained in low-glucose (1 g/L) Dulbecco’s modified Eagle medium (DMEM, PAA Laboratories GmbH, Pasching, Austria) supplemented with 10% fetal bovine serum (FBS, Biochrom AG, Berlin, Germany). Breast cancer cell lines were cultured in high-glucose (4.5 g/L) Dulbecco’s modified Eagle medium (DMEM, PAA Laboratories GmbH) supplemented with 10% fetal bovine serum (FBS, Biochrom AG). Both culture media were supplemented with 2 mM glutamine (PAA Laboratories GmbH), 10.000 IU/mL penicillin (Biotica, Part. Lupca, Slovakia), 5 μg/mL streptomycin (PAA Laboratories GmbH) and 2.5 μg/mL amphotericin B (Sigma-Aldrich, Taufkirchen, Germany). The cells were maintained at 37 °C in humidified atmosphere and 5% CO_2_, and the MSCs were then expanded and used for experiments not exceeding the 10th passage.

Human mammary gland adenocarcinoma cell line MDA-MB-231 (ATCC^®^ Number: HTB-26™), T47D (ATCC^®^ HTB-133™) and JIMT-1 (DSMZ no.: ACC 589) were purchased from stated sources and their identity was confirmed by STR profiling in July 2018. The cells were transduced with IncuCyte^®^ NucLight Lentivirus Reagents (Essen BioScience, Ann Arbor, MI, USA) to express nuclear red fluorescent protein (mKate2) according to manufacturer protocol. These cells are referred to as NLR-T47D, NLR-MDA-MB-231 and NLR-JIMT-1 (in manuscript shortened to NLR-T47D, NLR-MDA231 and NLR-JIMT). 

### 2.2. MSC Differentiation

Adipogenic differentiation was evaluated in MSCs plated at 3500 cells/well density in 96-well plates and maintained in low-glucose (1 g/L) DMEM medium supplemented with 60 μM indomethacin, 0.5 mM isobutylmethylxanthine, 0.5 μM hydrocortisone and 10% fetal bovine serum, GlutaMAX and antibiotic-antimycotic mix. The medium was changed every 2–3 days. The cells were washed with PBS after the 21st day of culture, fixed in 10% formalin and stained with Oil Red O (Sigma-Aldrich) for 2–5 min. The presence of adipocytes was detected by red stained lipid droplets. Osteogenic and chondrogenic differentiation was performed by StemPro Differentiation Kit (Gibco, Life Sciences, Carlsbad, CA, USA), where osteogenic differentiation was confirmed by detection of red stained calcium deposits using Alizarin Red S (Sigma-Aldrich) and chondrogenic positivity was proven by blue stained proteoglycans synthesized by chondrocytes using Alcian Blue stain (Sigma-Aldrich). Finally, the MSCs maintained in standard culture medium served as controls.

### 2.3. Immunophenotype

The identification and phenotyping of cultured MSCs was based on the defined International Society for Cellular Therapy (ISCT) standards and the use of human MSC Phenotyping Kit (Miltenyi Biotec, Bergisch Gladbach, Germany) [20]. The expression of CD90, CD105, CD14, CD20, CD34 and CD45 was assessed by BD FACSCanto™ II Flow cytometer (Becton Dickinson, USA) equipped with the FacsDiva program, and the data were then analyzed by FCS Express program.

### 2.4. Morphology and Wound Healing Assay

For MSC morphology analysis, 5 × 10^3^ MSCs in passage 2–4 were seeded in a 96-well plate and captured by IncuCyte ZOOM™ kinetic imaging system over 72-h period. For immunofluorescent analysis, cells were seeded on slides and, after reaching desired confluence, they were fixed with 4% PFA for 15 min, washed three times in PBS and subsequently stained with Actin-AF488 (1:500, diluted in ROTI) for 1 h at 37 °C and then with DAPI (1:500) for 15 min at 37 °C to stain the nuclei. MSC migration was evaluated in 96-well plates, where 23 × 10^3^ MSCs per well were seeded and analyzed by IncuCyte^®^ Scratch Wound Cell Migration and Invasion System and documented by the IncuCyte ZOOM™ kinetic imaging system. To assess the invasion capacity, 1 × 10^4^ MSCs + 2 × 10^4^ NLR-MDA231/NLR-T47D and 1.5 × 10^4^ MSCs + 3 × 10^4^ NLR-JIMT were seeded on ECM-coated 96-well plates and, after executing the scratch wound, they were immediately covered with 50% ECM (Matrigel, Sigma-Aldrich). MSCs were stained with Vybrant™ CFDA SE Cell Tracer Kit (Thermo-Fisher Scientific, Waltham, MA, USA) according to the manufacturer protocol. 

### 2.5. Evaluation of Proliferation

MSC doubling-time was evaluated by RealTime-Glo™ MT Cell Viability Assay (Promega Corporation, Madison, WI, USA). This was performed in a 96-well plate with 5 × 10^3^ cells per well seeded in eight replications. Relative luminescence was determined on LUMIstar GALAXY reader (BMG Labtechnologies, Germany) approximately every 12 h for 3 days. The luminescence values were extrapolated by a graph and trend line equation, and the mean luminescence after 24 and 48 h was determined by formula. The doubling-time was calculated as follows: doubling time = duration × log (2)/log (luminescence value after 48 h) − log (luminescence value after 24 h). The experiments were repeated at least twice with similar results, and the mean results were reported. 

Fluorescently labeled tumor cells and MSCs were mixed in a 2:1 ratio for co-culture experiments; 4 × 10^3^ NLR-JIMT and 2 × 10^3^ MSC or 2 × 10^3^ NLR-MDA231 and 1 × 10^3^ MSC were seeded in standard culture medium in 96-well plates. Each well was imaged every two hours by IncuCyte ZOOM™ Kinetic Imaging System (Essen BioScience, Newark Close, UK) until cells reached confluence. The tumor cell number was evaluated by IncuCyte ZOOM™ software (Essen BioScience) based on the enumeration of tumor cells’ red nuclei by kinetic imaging scanning. The values are expressed as means of replicates ± SD. Three-dimensional multicellular spheroids were prepared by seeding 2 × 10^3^ tumor cells mixed with 1 × 10^3^ MSCs in 96-well ultra-low attachment plates (Corning 7007, Corning Inc., Corning, NY, USA) in 100 µL of culture medium. Representative pictures of spheroids were taken after 7 days of culture by IncuCyte ZOOM™ Kinetic Imaging System (Essen BioScience) and the relative luminescence of spheroids was evaluated by the CellTiter-Glo™ 3D Cell Viability Assay (Promega Corporation).

### 2.6. Gene Expression Array

The Human Mesenchymal Stem Cells RT2 Profiler™ PCR Array then analyzed specific human mesenchymal stem cell gene expression in individual MSC groups (PAHS-082ZD; Qiagen, Hilden, Germany). RNA from 5 × 10^5^ MSCs was isolated by AllPrep RNA/Protein kit (Qiagen) and reverse transcribed with RT2 First Strand Kit (Qiagen). The expression of 84 human MSC-related genes was analyzed by RT2 SYBR Green Mastermix (Qiagen) and Bio-Rad CFX96™ Real-Time PCR Detection system (Bio-Rad Laboratories Ltd, Watford, UK). The CT cut-off was set at 35, and targets expressed at very low levels or undetected in MSC-H were excluded from relative expression calculations. The expression profile of MSC-H was used as a reference. Relative expression exceeding 4-fold alteration in tested samples was considered for further analysis.

### 2.7. Proteome Profiler

Proteome profile analysis of human cytokines and chemokines was performed with the XL Cytokine Array Kit (R&D Systems™, Minneapolis, MN, USA). ImageJ software (NIH, Bethesda, MD, USA) was used for the quantitative evaluation; the pixel density was determined and calculated. Serum-free conditioned media obtained from 2 × 10^5^ MSC-H or MSC-CA were loaded on the membranes with blotted antibodies and evaluated as recommended by the manufacturer.

### 2.8. Enzyme-Linked Immunosorbent Assay (ELISA)

IGF1 and leptin levels were quantified in the conditioned media from 1 × 10^5^ MSCs using a quantitative sandwich ELISA kit (Fine Biotech, China). The PTX3 level was quantified in the same way from the obtained media using a quantitative sandwich ELISA kit (R&D Systems). 

### 2.9. In Vivo Experiments

Six-week-old female SCID/Beige mice from SCID/bg, Charles River in Germany were used in accordance with institutional guidelines and approved protocols. The animals were bilaterally subcutaneously injected with a mixture of 5 × 10^5^ MSCs and 1x10^6^ NLR-JIMT cells re-suspended in 100 μL serum-free DMEM diluted 1:1 with ECM gel (Sigma-Aldrich). The animals were divided into five groups according to the type of injected MSC: control group of NLR-JIMT alone (*n* = 6), MSC-H (*n* = 6), MSC-DCIS (*n* = 6), MSC-CA (*n* = 6) and MSC-BRCA+ (*n* = 5). Alternatively, a mixture of 5 × 10^5^ NLR-JIMT cells and 2.5 × 10^5^ MSCs in 100 μL serum-free DMEM diluted 1:1 with ECM gel (Sigma-Aldrich) was injected bilaterally into the mammary fat pad of SCID/Beige mice. The animals were also divided into five groups according to the type of injected MSC: control group of NLR-JIMT alone (*n* = 3), MSC-H (*n* = 3), MSC-CA (*n* = 3), MSC-BRCA+ (2) (*n* = 3) obtained from breast tissue where prophylactic mastectomy was performed and MSC-BRCA+ (1) (*n* = 3) from contralateral breast of the same patient with confirmed relapsed invasive ductal carcinoma (pT1bpNx). The animals were regularly inspected for tumor growth, and tumor volume was calculated according to the formula: volume = (length × width2)/2. The animals were sacrificed according to the ethical guidelines when the tumor volume exceeded 1 cm^3^. The tumors were analyzed histologically and immuno-histochemically as described below.

All in vivo experiments were performed in the authorized animal facility under license No. SK UCH 02017 and approved by the institutional ethic committee and by the national competent authority of the State Veterinary and Food Administration of the Slovak Republic (Registration Number Ro:1976/17-221) in compliance with Directive 2010/63/EU of the European Parliament and the European Council and Regulation 377/2012 for the protection of animals used for scientific purposes.

### 2.10. The Immunofluorescence Analysis of Fresh Cryosections 

Tissues were embedded in Tissue-Tek (Sakura Finetek Europe, Alphen aan den Rijn, Netherlands), snap-frozen on dry ice and then cut into 10 μm sections on cryostat. The sections were gently washed three times with phosphate-buffered saline (PBS), fixed with 4% PFA for 15 min, washed three times in PBS and then permeabilized with 0.05% Triton X-100 in PBS for 15 min. The washed sections were incubated with ROTI protein free blocking solution (Carl Roth, Germany) for 30 min at 37 °C. Staining was performed by incubation with primary antibody Actin-AF488 (1:500, diluted in ROTI) for 1 h at 37 °C and then with DAPI (1:500) for 15 min at 37 °C to stain the nuclei. Finally, the slides were washed three times with PBS, and Fluormount-G^®^ medium (SouthernBiotech, Birmingham, AL, USA) was used to mount the coverslips. The staining patterns were analyzed using a Zeiss fluorescent microscope and automated imaging Metafer (MetaSystems GmbH, Altlussheim, Germany) (630× magnification).

### 2.11. Immunohistochemistry

Formalin-fixed, paraffin-embedded tumor tissues were cut into 5-µm sections. De-parafinization, rehydration and epitope retrieval via Target Retrieval solution high-pH (DAKO, Carpinteria, CA, USA) were performed under PT Link (Pre-Treatment module for tissue Specimens, DAKO) at 96 °C for 20 min. The slides were then washed in FLEX wash buffer (Tris-buffered saline solution containing Tween 20, pH 7.6) prior to loading onto the automated DAKO Autostainer_Link 48. Endogenous peroxidase was blocked by 5 min of incubation with FLEX peroxidase Block (DAKO), and sections were then incubated with primary antibodies, anti-human Ki67 MIB-1, anti-human Vimentin or anti-human smooth muscle actin (αSMA) for 20 min at RT (FLEX, DAKO). This was followed by incubation with LSAB2 System-HRP, Biotinylated Link for 15 min and then Strepatavidin-HRP for 15 min. Positive staining was visualized by the brown color from 3.3′-Diaminobenzidine (DAB substrate-chromogen solution, DAKO) after 5 min, and counterstaining was performed with hematoxylin (FLEX, DAKO) for 5–8 min. Sections with DAB-evident double-staining were incubated with other primary antibodies (Vimentin or αSMA) for 20 min, and in the same manner with LSAB2 system-HRP. Positive staining was visualized by Magenta (EnVision FLEX HRP Magenta Substrate Chromogen, DAKO) for 8 min (red color) and final sample counterstaining was with hematoxylin. The slides were washed between each incubation with 1× FLEX wash buffer and dehydrated by washing in alcohol, aceton:xylen (1:1) and xylen, each for 10 min. The slides were mounted with Q-D media (Bamed, Czech Republic) and the staining patterns were analyzed by Axio Scope A1, Zeiss microscope with Axiocam 105 color.

## 3. Results

### 3.1. Characterization of Mesenchymal Stromal Cells

Herein, we isolated mesenchymal stromal cells from breast adipose tissue and assigned each isolate to one of the following four groups schematically depicted in Figure 1: breast adipose tissue-derived MSCs from healthy donors (MSC-H), MSCs derived from breast adipose tissue from patients with pre-malignant lesions (MSC-DCIS) and adipose-derived MSCs from tissue adjacent to malignant lesions (MSC-CA and MSC-BRCA+). 

Each MSC isolate fulfilled the essential minimum criteria for multipotent stromal cells. MSCs were positive for CD90, CD105 and CD73 (>95%), but did not express CD14, CD20, CD34 and CD45 markers, as expected (<5% positive cells, representative sample in Figure 2A). Some MSC isolates were subjected to in vitro differentiation assay to confirm multi-lineage differentiation potential. MSCs readily differentiated into adipocytes (red stained lipid droplets), osteocytes (red stained calcium deposits) and chondrocytes (blue stained proteoglycans) under the in vitro culture conditions depicted in Figure 2B. Each MSC isolate produced actively proliferating cells which adhered to the plastic surface, and all isolates had fibroblast-like spindle-shape morphology. The phase-contrast photographs and actin immunofluorescence staining obtained 72 h after seeding revealed no morphological difference in the MSC-H, MSC-DCIS and MSC-CA groups (Figure 2C left) and also no age-dependent changes (Figure 2C right).

We also compared proliferation rates for individual MSC isolates and correlated these with patient age, BMI and diagnosis. Even though no significant difference was noted in age-related proliferation, a slightly lower proliferation trend was observed in the older MCS donors (Figure 3A left). However, the doubling-time of MSC-CA was significantly higher when compared to MSC-H (Figure 3A middle, *p* < 0.05, Mann Whitney test). This could be explained by the lower proliferation trend in older patients when solely MSC-CA doubling time was analyzed (Figure 3A right). It is well known that MSCs are endowed with the capacity to migrate towards tumors or the site of injury. Therefore, to see if this MSC trait could also be affected by tumor proximity, we compared the MSC migration in a standard wound healing assay. A wounded MSC monolayer was regularly imaged by live-cell kinetic imaging system and relative wound density was determined by wound confluence after 24 h. All MSC isolates showed high migratory capacity, with average 24-h wound confluence of 69% for 30–49-year-olds and 62% for those over 50. The migratory process, however, had no significant correlation with patient diagnosis (Figure 3B left) or age (Figure 3B middle). A moderately better wound healing process was evidenced in the MSC-DCIS, MSC-CA or MSC-BRCA+ of older patients (Figure 3B right). 

In an aspiration to analyze the expression profile of mesenchymal stromal cells derived from different origin, RT^2^ Profiler™ PCR array for human mesenchymal stem cell expression profiles was conducted on representative isolates from each group. The common alterations were noted in the MSC-DCIS and MSC-CA expression profiles (Figure 4A). These included; (1) up-regulation of brain-derived neurotrophic factor (*BDNF*), neurogenic locus notch homolog protein 1 (*NOTCH1*) and cytoskeletal Vimentin and (2) down-regulation of growth differentiation factor 15 (*GDF15*), insulin-like growth factor 1 (*IGF1*), matrix metallopeptidase 2 (*MMP2*), platelet-derived growth factor receptor β (*PDGFRB*) and transforming growth factor β3 (*TGFB3*). In addition, the MSC-BRCA+ cells exhibited down-regulation of Bone morphogenic protein 4 (*BMP4*) and up-regulation of SRY-box 9 (*SOX9*) and vascular cell adhesion molecule 1 (*VCAM1*). To examine how the MSC isolates within each group resemble each other in terms of expression profile, two different patient isolates from each group were compared. While the expression of mesenchymal stem cell markers in healthy donors was similar, with only a few genes being expressed differentially (Figure 4B left), the expression profiles in the MSC-CA (Figure 4B middle) and MSC-BRCA+ groups (Figure 4B right) were considerably different. This suggests that not only adjacent stroma but distant adipose tissue is affected by the presence of tumor mass as well and these changes remain in the MSCs even after the tumor-secreted factors are no longer present, as suggested by the altered expression profiles even after a certain time of culturing. However, if these tumor-caused changes are permanently retained in MSCs, such altered MSCs may later contribute to (or even cause) tumor recurrence. 

Further, we performed ELISA assays on more isolates to prevent misleading results caused by differences between individual MSC isolates. We have shown significantly lower concentrations of IGF1 in MSC-CA and MSC-BRCA+ (Figure 5A). As the molecular interplay between IGF1 and leptin, as well as its association with the pathogenesis of breast cancer, was shown previously, we have also analyzed the level of leptin which was significantly decreased in the group of MSC-CA (Figure 5B). After observing that the secretion of analyzed factors was decreased in cancer patient-derived MSCs, we were intrigued to see whether MSC-CA generally release less cytokines compared to MSC-H. The relative levels of human cytokines and chemokines in MSC culture medium were determined via a cytokine array kit. We were able to detect only nine out of 105 cytokines in MSC-CA media after 48-h culture compared to 20 out of 105 in MSC-H (Figure 5C). To see if this halted cytokine production could be connected to the tumor-released factors present in the tumor-adjacent adipose tissue used for MSC-CA isolation, we cultured healthy MSCs in NLR-MDA231 conditioned medium (CM-BCC) for 2 weeks. The analysis of the exemplary cytokine PTX3 concentration released to the media showed that it was decreased in MSC-H cultured in CM-BCC compared to the same isolate cultured in standard growth control media (Figure 5D). 

### 3.2. MSC Interactions with Breast Cancer Cells

After seeing that the cytokine production in MSC-CA is seemingly halted, we speculated different ways of communication between MSCs and tumor cells. We looked at the presence of direct cell-to-cell contacts in MSC-breast cancer cell co-culture and observed thin plasma membrane structures formed between cancer cells and MSC, which could allow cellular cross-talk leading to alteration of cell properties (Figure 6A). The altered functions and phenotype of breast cancer cells were analyzed in co-culture with MSCs of different origins. Direct co-culture of MSCs with breast cancer cells resulted in a more mesenchymal-like morphology of NLR-T47D and NLR-JIMT cells (Figure 6B) and their proliferation, regardless of the co-cultured MSC origin, was increased (Figure 6C middle, right). The NLR-MDA231 cell proliferation in vitro, however, was not affected by direct contact with MSCs (Figure 6C left).

To test the MSCs’ effect on NLR-JIMT and NLR-T47D cell proliferation in a more relevant in vitro model, we proceeded to co-culture in 3D non-adherent culture conditions (Figure 7A). Although in NLR-T47D co-culture the spheroid size was bigger, the structure was less compact, and the luminescent assay showed significantly less ATP in all MSC groups (Figure 7B left). NLR-JIMT-MSC co-culture in 3D conditions showed MSC support of tumor cell proliferation, and this was also confirmed by luminescent assay (Figure 7B right). While this leads to the assumption that the MSC-mediated augmentation of cancer cell proliferation is cell-line specific, there was no difference in effect regarding whether the MSCs were isolated from the adipose tissue of healthy donors or cancer patients. To gain deeper insight into how the MSCs affect breast tumor cells, the invasion profiles of both MSC populations in co-culture with tumor cells were evaluated in IncuCyte^®^ Scratch Wound Invasion assay. Interestingly, MSC-CA invaded the 50% ECM much more rapidly than MSC-H and, moreover, they augmented the invasion of NLR-MDA231 cells as well (Figure 7C).

### 3.3. The Effect of MSCs on Tumor Growth In Vivo

To evaluate the effects of MSCs on tumorigenicity in vivo, immuno-compromised SCID/Beige mice were subcutaneously injected solely with NLR-JIMT cells or with NLR-JIMT cells mixed with MSCs from different origins (Figure 8A). The NLR-JIMT cell line represents the HER2-enriched non-luminal tumor type characterized by aggressive growth with intermediate prognosis. The confirmed insensitivity to HER-2 inhibiting drugs (trastuzumab and pertuzumab) makes it a valuable experimental model for studies of resistance mechanisms. Subsequent tumor volume analysis on Day 15 revealed the supportive effect of MSCs on tumor growth in the co-injected xenografts compared to those composed solely of tumor cells (Figure 8B). In addition, the NLR-JIMT co-injected with cancer patient-derived MSC-BRCA+, MSC-DCIS and MSC-CA produced a significantly higher tumor volume than NLR-JIMT co-injected with healthy donor-derived MSC-H (*p* < 0.01 in MSC-H vs. MSC-CA and MSC-H vs. MSC-BRCA+; *p* < 0.05 in MSC-H vs. MSC-DCIS). The highest pro-tumorigenic effect, however, was observed in xenografts formed by NLR-JIMT cells co-injected with invasive cancer patient-derived MSC-CA. When the NLR-JIMT were co-injected with MSC-CA, the average tumor volume on Day 15 was 338.3 mm^3^ compared to 76.6 mm^3^ volume of the MSC-H co-injected tumors. This demonstrates a striking MSC-CA pro-tumorigenic effect. The experimental mice were sacrificed when tumors reached more than 11 mm in any dimension. Subsequent histological analysis of the tumor xenografts showed decreased Ki67 expression in the center of xenografts formed by NLR-JIMT co-injected with cancer patient-derived MSC-BRCA+, MSC-DCIS and MSC-CA than in NLR-JIMT co-injected with healthy donor-derived MSC-H, or NLR-JIMT alone (Figure 8C, 1st column). This is explained by MSC localization in the xenograft center, and the subsequent denser micro-environment surrounding the tumor cells.

The αSMA and Vimentin staining in Figure 8C suggests that in tumors injected with cancer patient-derived MSCs, and particularly in those with MSC-BRCA+, the MSCs attempt to form aligned structures around the tumor cells that resemble cellular pathways. α-SMA was used to visualize the localization of MSCs within the tumor xenografts, and although human αSMA antibody was used, the mouse endothelial cells in xenografts formed by NLR-JIMT cells were stained as well. Therefore, additional staining with the MSC marker Vimentin was performed, and confirmed that the αSMA positive cells were indeed MSCs. This was independently evaluated by a pathologist (MA) and histology specialist (MBu) who confirmed specific MSC-positive staining in the carcinoma-associated MSC-containing xenografts. In addition, Figure 8D confirms that Ki67 staining of the xenograft periphery depicts clusters of tumor cells invading surrounding tissue in NLR-JIMT co-injected with MSC-H, MSC-BRCA and MSC-DCIS xenografts. This manifest substantial effect of these MSCs on tumor cell invasiveness and also highlights collective cell-scattering, where the tumor cells detach and spread in small clusters. In contrast, the histological analysis of tumors formed by NLR-JIMT co-injected with MSC-CA showed collective cell migration with a distinguishable invasive front (Figure 8E; indicated with arrow). These tumor cells were in immediate proximity to adipocytes and spread along adipocyte intercellular spaces. Moreover, the adipocytes closest to the tumor invasive front are clearly smaller and have wider intercellular spaces (Figure 8E; indicated with asterisks). Finally, Vimentin staining in these tumor cells suggested epithelial-to-mesenchymal transition (EMT). Vimentin staining was positive in the tumor cell cytoplasm of xenografts co-injected with MSC-CA located in the invasive front (Figure 8F). Ki67 staining quantification revealed its uneven distribution throughout the tissue section, but its expression in all groups proved similar when a greater number of zones was investigated (Table 1).

In order to verify the observed effects in a clinically more relevant model, an orthotopic model of mammary carcinogenesis was used (Figure 9A). The mixture of NLR-JIMT and MSC-CA resulted in a higher tumor volume in the mammary glands of injected mice compared to the mixture of NLR-JIMT and MSC-H. The co-injection of tumor cells with MSC-BRCA (2) obtained from patient breast tissue where prophylactic mastectomy was performed and MSC-BRCA (1) from contralateral breast of the same patient with confirmed relapsed invasive ductal carcinoma (pT1bpNx) demonstrated that MSCs isolated from the adipose tissue adjacent to a tumor mass have a more aggressive phenotype and augmented tumor volume in NLR-JIMT + MSC-BRCA (1) (Figure 9B). Ki67 staining of the xenograft periphery confirmed the invasive character of tumor cells co-injected with MSCs isolated from tumor-adjacent adipose tissue. Interestingly, the increased metastatic potential in the presence of MSC-CA was confirmed also by the increased innervation of tumor xenografts. The perineural and intraneural invasion of CK7- and vimentin-positive tumor cells was detected (Figure 9C), as well as the presence of tumor cells inside the blood vessels (Figure 9D).

## 4. Discussion

Tumor–stroma interaction, and particularly the cytokine–cytokine receptor interaction pathway, is significantly altered in highly aggressive tumors [21]. While many studies have evaluated the role of mesenchymal stromal cells in the tumor microenvironment [22], the majority were based on adipose MSCs from non-breast origins [23,24]. In contrast, we compared normal breast adipose tissue-derived MSCs and their tumor-activated counterparts and revealed phenotypic, molecular and functional changes which determined the stromal cell activation status.

There is a growing substantial interest in identifying MSC proliferation properties for future application in cell therapy and tissue engineering [25,26]. Herein, the growth kinetics and population doubling-time were influenced by age when younger and older donors were compared, as shown by others [27,28]. While the correlation between MSC doubling-time and age or BMI status in MSC donors was not significant at first sight, the separate analysis of doubling-time in younger and older patient groups for healthy and cancer patient-derived MSCs highlighted an increased doubling-time in the older group. Apparently, the conflicting observations are likely related to increased doubling-time in MSC-CA compared to MSC-H. 

MSC morphological analysis did not reveal any changes related to diagnosis or age. The changes might be evident only when MSCs differentiate into CAF in the tumor micro-environment [10], while adjacent adipose tissue-derived MSC morphology remains unaffected or the tumor cell influence in the culture is lacking.

In addition to doubling-time changes, we also identified altered gene expression and cytokine production in breast cancer-derived MSCs. The most evident was the down-regulation of *GDF5*, *GDF6*, *IGF1*, *PDGFRB* and *TGFB3* genes in MSC-CA compared to MSC-H. Morales et al. [21] identified 76 differentially expressed genes associated with the metastasis of relapsed primary tumors. Two of these, down-regulated *GDF5* and *TGFB3*, corresponded to the results observed in our analysis. *NOTCH1* expression has also been implicated in cancer cell metastasis, and breast cancer patients positive for *NOTCH1* have experienced shorter disease-free survival [29,30]. Here, it is most likely that adipose tissue’s close proximity to a breast tumor correlates with the observed MSC expression changes, and these could well explain the increased in vivo invasion of NLR-JIMT + MSC-CA tumor xenografts. 

It was reported that BMP4 possesses both tumor-suppressive and oncogenic properties in breast cancer and that it is a potent suppressor of breast cancer metastasis [31,32]. Therefore, the decreased *BMP4* expression in MSC-BRCA+ could contribute to the in vivo cell cluster migration we observed in the more distant parts of tumors. In addition, BMP4 down-regulation in the tumor micro-environment could lead to the formation of these clusters and thus increase tumor cell metastatic ability. Furthermore, increased *SOX9* expression in MSC-BRCA+ correlates with prognostic significance in invasive ductal carcinoma and metastatic breast cancer [33]. Therefore, we assume it can also be associated with the observed invasive nature of NLR-JIMT + MSC-BRCA+ tumors. We also suggest that the up-regulation of brain-derived neurotrophic factor (BDNF), which supports innervation in tumor xenografts, could be a promising determinant of tumor cell invasion and metastasis. We propose that mesenchymal stromal cells in the tumor proximity are pushed by tumor cells to help in nerve recruitment and, interestingly, these alterations in MSCs become permanent as they remain even when the tumor is not present anymore.

In MSC-CA and MSC-DCIS, here we determined the decreased expression of *GDF15* and *IGF*, but this conflicts with published results which show increased expression of these genes in the tumor micro-environment [34]. 

We propose that the decreased cytokine production by MSC-CA could correlate with tumor presence in adipose tissue, as also healthy MSCs cultured in NLR-MDA231 conditioned media showed altered cytokine production. However, tumor cells co-cultured with MSC-CA exhibited a more aggressive phenotype in vitro and in vivo, which could be associated with direct cell-to-cell communication and connections, indicating the presence of nanotubes in MSC–tumor cell cultures. Moreover, histological tumor xenograft analysis revealed functional changes associated with epithelial-to-mesenchymal transition induction and increased tumor cell invasion in co-culture with MSC-CA. While the collective migration of tumor cells was combined with high Ki67 and VIM positivity in the invasive front, the cell invasion differed in tumor cells co-injected with MSC-DCIS compared to MSC-CA. This identifies that MSC-CA have a more aggressive phenotype.

Vimentin up-regulation is an EMT-specific marker of increased cancer cell motility and migration [35,36]; therefore, its positive staining in NLR-JIMT-MSC-CA xenograft suggests significant functional changes in cancer patients’ MSCs where breast adipose tissue has close proximity to the tumor. The functional changes are also suggested by the smaller adipocytes and wider intercellular spaces in close proximity to the xenograft invasive front, suggesting additional extracellular matrix alterations [37]. 

Finally, the combined data provide the conclusion that MSCs in cancer patient adipose tissue have inherently altered expression profiles and functional characteristics which enhance their ability to support in vivo tumor cell propagation. Moreover, the adipose tissue MSCs derived from closely adjacent tumor tissue increased the volume of tumor xenografts when co-injected with NLR-JIMT subcutaneously and orthotopically, and also supported the release of tumor cell clusters. Orthotopic model confirmed the pro-tumorigenic phenotype of MSC-CA which induced the perineural invasion of tumor cells. Our data suggest the capacity of cancer patient-derived MSCs to support cell scattering and invasion by involving tumor cell epithelial-to-mesenchymal transition. 

## 5. Conclusions

To the best of our knowledge, we have described permanent proliferative and functional changes in tumor-adjacent adipose tissue-derived MSCs for the first time. The analysis of MSCs isolated from breast cancer patients provides improved understanding of the changes provoked in adipose tissue by close proximity to the tumor and also identifies MSCs’ role in the promotion and progression of tumor growth. 

Herein, we showed that the micro-environment of adipose tissue closely adjacent to breast tumor tissue is composed of tumor-exposed MSCs which differ in doubling-time, expression profile, cytokine production and tumor-promoting ability shown by perineural invasion in vivo compared to the MSCs in healthy adipose tissue. Further study focused on the specific molecular pathways responsible for the activation and re-programming of MSCs exposed to tumor micro-environment should reveal potential therapeutic strategies which will block the tumor-induced alterations caused to adjacent adipose tissue.

## Figures and Tables

**Figure 1 cells-09-00480-f001:**
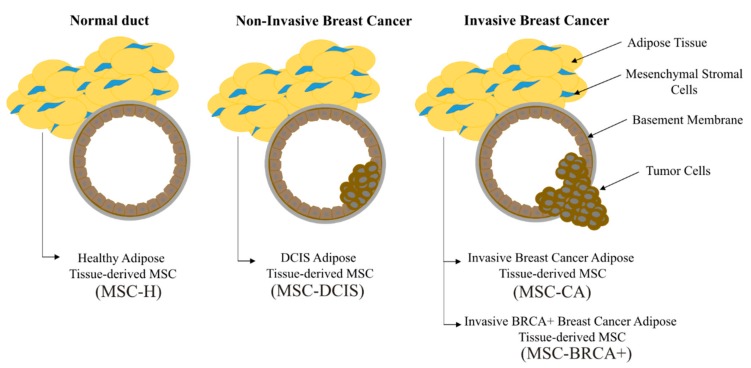
Mesenchymal stromal cells (MSCs) were isolated from four breast adipose tissue origins; healthy donors (MSC-H), breast cancer patients with non-invasive (MSC-DCIS) or invasive tumors (MSC-CA), and BRCA+ breast cancer patients (MSC-BRCA+). In breast cancer patients, adipose tissue at a distance of 1.5-2 cm from the tumor was used.

**Figure 2 cells-09-00480-f002:**
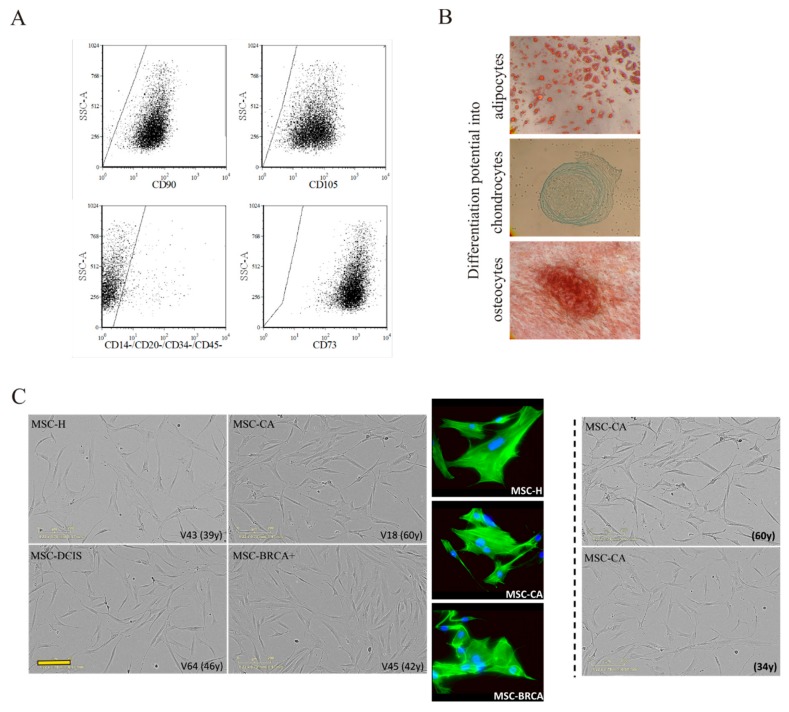
(**A**) MSC-specific marker expression was analyzed by flow cytometry. MSCs expressed CD73, CD90 and CD105 on their surface, but lacked hematopoietic/endothelial marker expression. (**B**) MSC multipotency was analyzed by their differentiation potential into adipocytes, osteocytes, and chondrocytes. Magnification 100×. (**C**) MSC phenotype was documented by IncuCyte ZOOM™ kinetic imaging system and actin immunofluorescent staining (actin - green, DAPI - blue). The morphology of MSCs obtained from donors with different diagnosis (left) or age (right) did not differ over the analyzed period. Scale bar 200 µm. One representative picture from each group is shown.

**Figure 3 cells-09-00480-f003:**
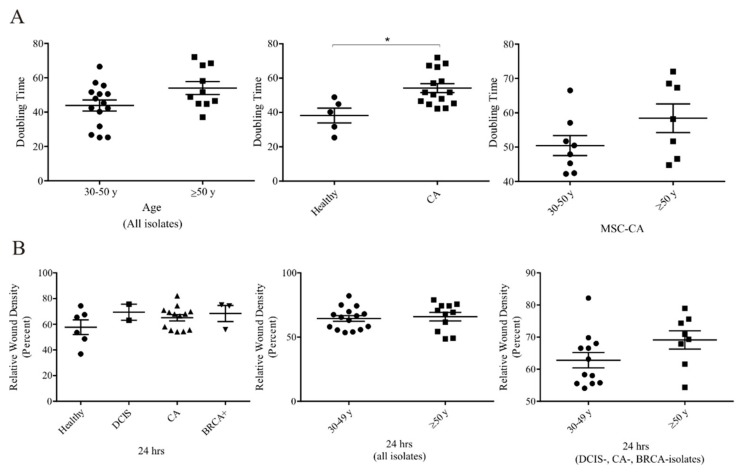
Functional changes in cancer patient adipose tissue-derived MSCs (**A**) Left: MSC doubling-time in all MSC isolates had increasing tendency in older donors, but this tendency was not statistically significant. Middle: Based on the diagnosis of MSC donors, we observed significantly longer doubling-time in MSCs isolated from patients with invasive cancer compared to healthy MSCs (* *p* < 0.05; Mann-Whitney test). Right: The trend of increased doubling-time was present also in the MSC-CA group. (**B**) MSC migration potential analysis based on diagnosis (left), age in all MSC groups (middle) or age in solely cancer patients (right) showed no statistically significant differences. MSCs used for analysis did not exceed 10th passage.

**Figure 4 cells-09-00480-f004:**
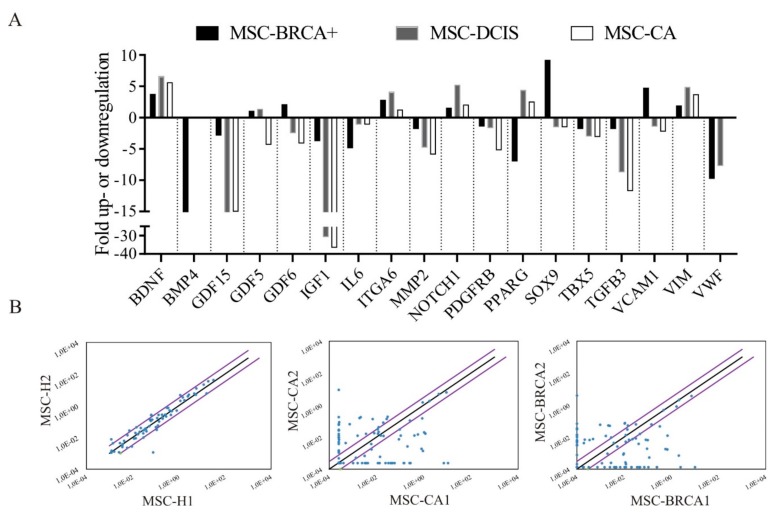
Expression profiles of healthy and cancer patient adipose tissue-derived MSCs. (**A**) RT^2^ Profiler™ PCR human mesenchymal stem cells array of individual MSC isolates used in in vivo study revealed several gene expression changes. (**B**) Scatter plot of mesenchymal stem cell gene expression at the mRNA level comparing two different healthy (left), cancer (middle) and BRCA+ isolates (right). Lateral diagonal lines indicate a 2-fold increase or decrease. Results were obtained using the RT2 Profiler PCR Array Data Analysis software (at Qiagen data analysis web portal). MSCs used for analysis did not exceed 10th passage.

**Figure 5 cells-09-00480-f005:**
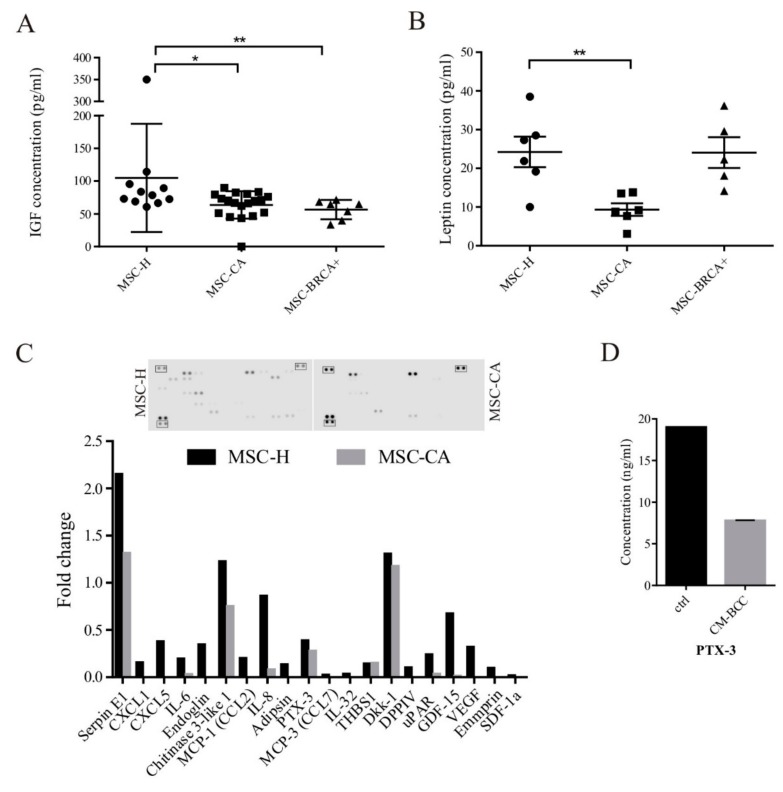
Secretion profile of cancer patient adipose tissue-derived MSCs is halted by tumor cell-secreted factors. (**A**) Decreased expression of IGF1 correlated with decreased IGF1 concentration detected in MSC-CA cell media (* *p* < 0.05; ** *p* < 0.01; Mann–Whitney test). (**B**) Leptin concentration in MSC-CA was also lower compared to MSC-H. The IGF1 and leptin concentration was measured by ELISA test in MSC medium after 48 h of culture (* *p* < 0.05; ** *p* < 0.01; Mann–Whitney test). (**C**) Cytokine analysis revealed a decreased release of cytokines and chemokines in MSC-CA isolate. The relative change in analyte level between MSC groups was determined by subtraction of each pair of capture antibody from the reference spot signal on the corresponding membrane. (**D**) The PTX3 concentration was decreased in healthy MSCs cultured for 2 weeks in NLR-MDA231 conditioned media. The concentration of PTX3 was measured by ELISA test in MSC medium after 48 h. MSCs used for analysis did not exceed 10th passage.

**Figure 6 cells-09-00480-f006:**
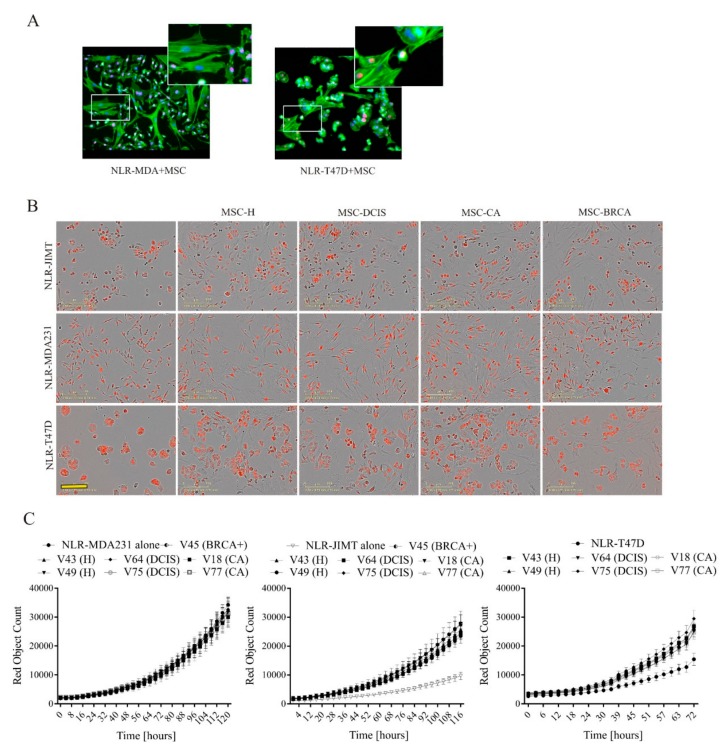
MSC–tumor cell interactions in 2D in vitro conditions. (**A**) Thin plasma membrane structures are formed between cancer cells and MSCs in co-culture allowing cell-to-cell communication and signaling. Magnification 200×. Cytoplasmic actin was stained green, nuclei were stained with DAPI (blue). The nuclei of tumor cells also expressed red fluorescent protein, therefore they appear as magenta colored. (**B**) MSC co-culture with tumor cells expressing red fluorescent nuclear protein resulted in more mesenchymal-like cell morphology of co-cultured NLR-T47D and NLR-JIMT cells. Scale bar: 200 µm. (**C**) Direct 7-day co-culture of NLR-JIMT and NLR-T47D breast cancer cells with MSCs of different origins highlighted the supportive role of MSCs in tumor cell proliferation, but no diagnosis-specific effects on proliferation were observed. NLR-MDA231 proliferation was not enhanced by the presence of MSCs.

**Figure 7 cells-09-00480-f007:**
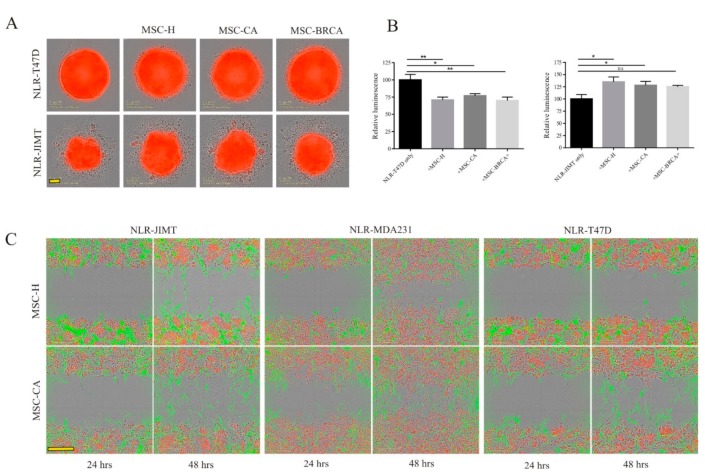
Faster invading cancer patient-derived MSCs are followed by more rapidly invading tumor cells. (**A**) Monoculture vs. co-culture in 3D non-adherent culture conditions showed less compact NLR-T47D-MSC spheroids and bigger NLR-JIMT-MSC spheroids, but no difference between MSC isolates was observed. Scale bar: 100 µm. Breast cancer cells—red color, MSCs—unstained. (**B**) Luminometric measurement of spheroid cultures after 7 days revealed significantly lower ATP amount in NLR-T47D-MSC co-culture and higher ATP amount in NLR-JIMT-MSC co-culture (* *p* < 0.05; ** *p* < 0.01; Mann–Whitney test). In the control group (only breast cancer cells without MSC), 8 samples were analyzed. In each group with MSC, 4 spheroids were analyzed. (**C**) MSC-CA exhibited increased invasion potential in Scratch wound invasion assay after 24 or 48 h. The invasion of tumor cells was also increased in co-culture with MSC-CA compared to MSC-H (1 × 10^4^ MSCs + 2 × 10^4^ NLR-MDA231/NLR-T47D and 1.5 × 10^4^ MSCs + 3 × 10^4^ NLR-JIMT were seeded on Matrigel coated 96-well plates and covered with 50% Matrigel). Scale bar: 200 µm. MSCs used for analysis did not exceed 10th passage. MSCs were stained with Vybrant™ CFDA SE Cell Tracer Kit (green color), breast cancer cell lines expressed red fluorescent protein.

**Figure 8 cells-09-00480-f008:**
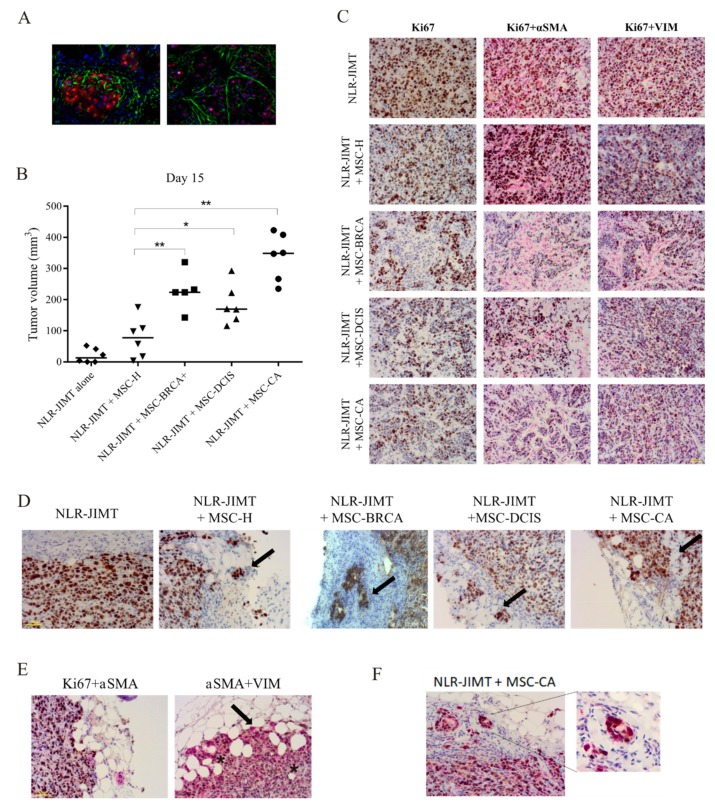
Cancer patient adipose tissue-derived MSCs showed a pro-tumorigenic effect on subcutaneous tumor xenografts in vivo. (**A**) Actin immunofluorescence staining of mice xenograft cryosection shows xenograft composition formed by tumor cells and MSCs (left—xenograft periphery, right—xenograft center; red—cancer cell nuclei, green—actin cytoplasm staining, blue - nuclei staining with DAPI. Magnification 630×.) (**B**) 5 × 10^5^ MSCs of different origin were subcutaneously co-injected with 1 × 10^6^ NLR-JIMT cells in immuno-compromised SCID/Beige mice. Tumor volume examination on Day 15 revealed profound supportive effect of MSCs on tumor growth in the co-injected xenografts. Xenografts composed solely of tumor cells failed to induce significant tumor volume in the analyzed period. Tumor volume was calculated by formula: volume = (length × width2)/2 (* *p* < 0.05, ** *p* < 0.01, Man-Whitney test). The significantly most supportive effect was observed in mice injected with NLR-JIMT + MSC-CA. (**C**) Detection of Ki67, αSMA and VIM markers by immuno-histochemistry in tumor tissue sections. Mice were sacrificed when the tumor xenograft reached 1 cm^3^. Xenografts were fixed with formaldehyde, embedded in paraffin and processed for immuno-histochemical staining with monoclonal antibodies. Representative images of tumors formed by NLR-JIMT co-injected with cancer patient-derived MSCs (BRCA+, DCIS, CA) showed lower Ki67 positivity in the tumor center than in NLR-JIMT co-injected with healthy donor-derived MSC-H. The αSMA and Vimentin (VIM) staining showed that mainly the MSC-CA and MSC-BRCA+ attempted to form aligned pathway-like structures around the tumor cells. Magnification 200×. (**D**) Representative Ki67-stained pictures of xenograft periphery showed clusters of tumor cells invading surrounding stroma in tumors formed by NLR-JIMT co-injected with MSCs (MSC-H, -DCIS, -BRCA) and collective cell migration with a distinguishable invasive front was observed in the group co-injected with MSC-CA. (**E**) Immuno-histochemical staining of αSMA and Vimentin revealed up-regulation in tumor cells located in the invasive front of the xenografts co-injected with MSC-CA. This suggests the epithelial-to-mesenchymal transition of tumor cells. Asterisks identify smaller adipocytes with dilated inter-cellular spaces near the tumor invasive front. (**F**) Detail of the xenograft periphery showing Vimentin positivity in tumor cells.

**Figure 9 cells-09-00480-f009:**
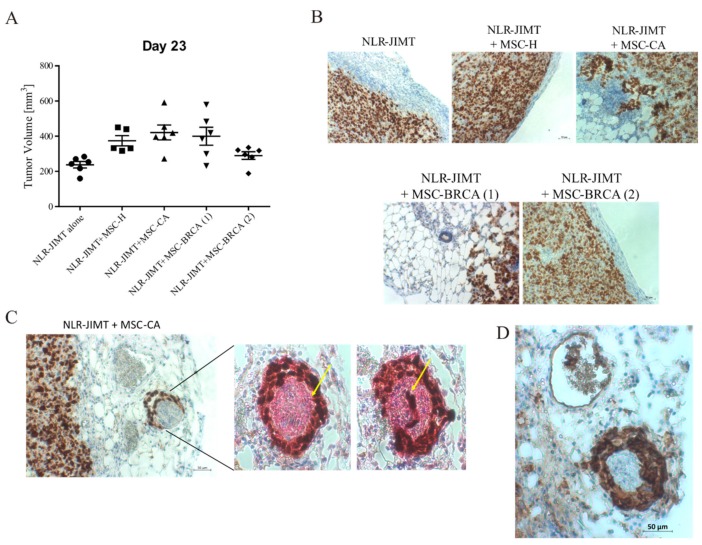
Pro-tumorigenic effect of MSC-CA in orthotopic mouse model. (**A**) Mixture of 5 × 10^5^ NLR-JIMT cells and 2.5 × 10^5^ MSCs in 100 μL serum-free DMEM diluted 1:1 with ECM gel (Sigma-Aldrich) was injected bilaterally into mammary fat pad of SCID/Beige mice. The animals were divided into five groups according to the type of injected MSC: control group of NLR-JIMT alone (*n* = 6), MSC-H (*n* = 6), MSC-CA (*n* = 6), MSC-BRCA (2) (*n* = 6) obtained from breast tissue where prophylactic mastectomy was performed and MSC-BRCA (1) (*n* = 6) from contralateral breast of the same patient with confirmed relapsed invasive ductal carcinoma. Tumor volume was calculated according to the formula: volume = (length × width2)/2. The animals were sacrificed when the tumor volume exceeded 1 cm^3^. The most supportive effect was observed in mice injected with NLR-JIMT + MSC-CA and MSC-BRCA (1). (**B**) Ki67 staining of xenograft periphery showed collective cell invasion in the group co-injected with MSC-CA. This manner of invasion was also observed in the MSC-BRCA (1), but was lacking in the MSC-BRCA (2) co-injected group. While both MSC-BRCA isolates come from the same patient, the former were isolated from a breast with relapsed ductal carcinoma and the later from a contralateral healthy breast where prophylactic mastectomy was performed. MSCs derived from breast adipose tissue with confirmed presence of tumor (MSC-BRCA (1)) increased the invasion of tumor cells in xenograft periphery. (**C**) Nerve fibers were detected in serial sections of NLR-JIMT + MSC-CA orthotopic xenografts using IHC staining with specific neuronal marker PGP9.5. The perineural and intraneural tumor cell invasion (yellow arrow pointing at invading single tumor cell in the left picture and group of tumor cells in the right picture) was present only in the MSC-CA group. (**D**) CK7 antibody (breast cancer marker) staining confirmed the presence of tumor cells in the perineural space and also inside the blood vessel.

**Table 1 cells-09-00480-t001:** Ki67 score in tumor xenografts. Ki67 expression analysis in several areas of tissue section revealed its uneven distribution throughout the xenograft. When a higher number of zones was investigated, the Ki67 staining quantification showed similar expression in all groups.

Injected Cells (NLR-JIMT)	Ki67 Positivity in Analyzed Area (%)	Average Ki67 Positivity (%)
**alone**	59.59	72.9
	66.53	
	92.46	
**+MSC-H**	77.55	70.9
	59.34	
	75.88	
**+MSC-BRCA**	76.77	73.4
	69.41	
	73.9	
**+MSC-DCIS**	95.35	86.1
	68.45	
	94.57	
**+MSC-CA**	64.95	72.5
	65.91	
	86.71	

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
