# Peer review of "Permanent Pro-Tumorigenic Shift in Adipose Tissue-Derived Mesenchymal Stromal Cells Induced by Breast Malignancy"

_cells, 2020, doi:10.3390/cells9020480_

Round 1

Reviewer 1 Report

In this manuscript, Plava and colleagues reported that MSCs derived from breast cancer patients are different from MSC derived from healthy donors in gene expression profiles and the ability to support tumor growth. The experiments were generally well-designed and it is an interesting topic. However, there are several points need to be addressed.

The authors claimed that MSCs derived from cancer patients had a permanent pro-tumorigenic shift due to the presence of tumors. However, it is also possible that the development of breast cancer was at least partially attributed to the altered gene expression in MSCs, which means the abnormal MSCs caused breast cancer.  It is not clear how the difference in MSCs gene expression contributed to their different pro-tumorigenic activity. The alteration of which cytokines/genes most likely promoted tumor growth? In vivo tumor growth study showed increased tumor growth in the presence of cancer patient-derived MSCs. However, Ki67 staining showed more abundant proliferating cells in the presence of healthy MSCs. Quantification of tumor cell and MSC proliferation is needed. Is there any difference in EMT gene expression in tumor cells in the presence of different types of MSCs?

Reviewer 2 Report

Dear colleagues!

After review of the manuscript by Plava et al. I have the following comments regarding the study and presentation of results. Overall, the study presents an original object and a good hypothesis regarding the proximity of MSC to tumour site (keeping in mind specific types of tumours) as a crucial aspect of a phenotype shift in MSC that favours cancer progress in later stages.

The paper requires a moderate level of English language proofing as far as certain sentences are very hard to perceive. I suggest a thorough check by an Editor or a native-speaking proficient writer.

Introduction is well-written and supplemented by a figure to understand the background for the study and general idea that continues in Fig.1

Regarding specific criticism I have found the following issues that I believe will require attention from the Authors prior to further handling of the manuscript.

1) In Fig. 4 authors provide an important graph in panel A. However, it is very hard to discern between genes and groups. The panel should be stretched to fit in the names of genes under corresponding bars. In actual design it is absolutely unreadable.

2) Line 330-331 provide an important claim that in MSC-CA authors detected 9/105 cytokines. However it is not clear how many were detectable in MSC-H? Probably, total protein measurement could be a supporting evidence for normalisation of data in quantitative array kit. Or the kit has a normalising dot - please, highlight it in the figure then.

3) Do bar-graphs in Fig. 5C present a densitometry of dot-array above them? Then it is a bit confusing as far as dots in array look brighter in MSC-CS while most detectable proteins in bar-graphs are downregulated compared to MSC-H.

4) Data provided in Fig. 6 is barely readable - probably this can be moved to Supplement with a better resolution and size. In current state it is not appropriate.

5) In experiments in Fig. 8 JIMT line was chosen. I understand why it was used - however, a short rationale can be put at ht beginning of the in vivo part to give a reader a better understanding of cell line choice. It should also be stressed that w/o MSC supplementation (even healthy ones) this cell line failed to induce a significant tumour volume in SCID mice

6) Lines 441-444 are very confusing. It is not clear how Authors discern between mouse endothelium and human tumours or MSC cells. Why would Authors rely on a-SMA as a marker of endothelium and not use anti-mouse CD31 in case this is an important finding. However, I would rather point the possibility of EndoMT in case you may detect mouse CD31/Vim+ cells. Please, explain this passage as far as it is extremely confusing.

7) Legend of Fig. 9 - is it correct that most supportive effect was in MSC-BRCA2 obtained from a formally non-cancerous breast compared to MSC-BRCA1 obtained from a peri-tumorous region? The sentence in Lines 494-496 increases confusion as far as it claims that only in MSC-BRCA1 invasion of tumour cells was observed? Please, explain.

8) In line 539 ";" sign after MSC-DCIS looks a typo error

9) Image data in Fig 7 C does not support the claim that "MSC-CA exhibited  increased invasion potential in Scratch wound invasion assay after 24 or 48 hours." Invasion in MSC-H and MSC-CA hardly looks different.

Yours, Reviewer.

Round 2

Reviewer 2 Report

Dear colleagues, I appreciate the changes made in the manuscript!

Yours, Reviewer.